# A versatile site-directed gene trap strategy to manipulate gene activity and control gene expression in *Caenorhabditis elegans*

**Haania Khan**[1¤], **Xinyu Huang**[1,2], **Vishnu Raj**[1], **Han Wang**[1,2]*

**1** Department of Integrative Biology, University of Wisconsin-Madison, Madison, Wisconsin, United States of America, **2** Genetics Training Program, University of Wisconsin-Madison, Madison, Wisconsin, United States of America

¤ Current address: Department of Biochemistry, University of Wisconsin-Madison, Madison, Wisconsin, United States of America
* han.wang@wisc.edu

**Data Availability Statement:** All relevant data are within the manuscript and its Supporting Information files.

## Abstract

The ability to manipulate gene activity and control transgene expression is essential to study gene function. While several genetic tools for modifying genes or controlling expression separately are available for *Caenorhabditis elegans*, there are no genetic approaches to generate mutations that simultaneously disrupt gene function and provide genetic access to the cells expressing the disrupted gene. To achieve this, we developed a versatile gene trap strategy based on cGAL, a GAL4-UAS bipartite expression system for *C. elegans*. We designed a cGAL gene trap cassette and used CRISPR/Cas9 to insert it into the target gene, creating a bicistronic operon that simultaneously expresses a truncated endogenous protein and the cGAL driver in the cells expressing the target gene. We demonstrate that our cGAL gene trap strategy robustly generated loss-of-function alleles. Combining the cGAL gene trap lines with different UAS effector strains allowed us to rescue the loss-of-function phenotype, observe the gene expression pattern, and manipulate cell activity spatiotemporally. We show that, by recombinase-mediated cassette exchange (RMCE) via microinjection or genetic crossing, the cGAL gene trap lines can be further engineered *in vivo* to easily swap cGAL with other bipartite expression systems' drivers, including QF/QF2, Tet-On/Tet-Off, and LexA, to generate new gene trap lines with different drivers at the same genomic locus. These drivers can be combined with their corresponding effectors for orthogonal transgenic control. Thus, our cGAL-based gene trap is versatile and represents a powerful genetic tool for gene function analysis in *C. elegans*, which will ultimately provide new insights into how genes in the genome control the biology of an organism.

## Author summary

Genetic tools are critical to understanding how genes function to control the biology of an organism. Here we use a bipartite expression system to develop the first gene trap strategy for the model organism *Caenorhabditis elegans*—a powerful and versatile genetic tool that

**Funding:** This work was supported by NIH (K99/R00GM126137 and R35GM150658 to H.W.), Whitehall Foundation (to H.W.), and startup funds from University of Wisconsin-Madison and the Wisconsin Alumni Research Foundation to (H.W.). https://www.nigms.nih.gov/ http://www.whitehall.org/ https://www.warf.org/ https://www.wisc.edu/ The funders had no role in study design, data collection and analysis, decision to publish, or preparation of the manuscript.

**Competing interests:** The authors have declared that no competing interests exist.

can specifically disrupt virtually any gene and precisely control transgene expression at the same time. Gene trap strains can be used to reveal endogenous expression patterns, perform genetic rescue experiments, and manipulate cell-specific activity; they can also be easily converted to gene traps with other bipartite expression systems. Our site-specific, robust, and swappable gene trap strategy will greatly facilitate genetic studies in *C. elegans* and can potentially be applied to other model organisms due to its versatility.

## Introduction

The biology of an organism, such as its development, physiology, and behavior, is specified by its genome. These biological processes require genes encoded in the genome to act in different cells at specific times. A powerful way to infer gene function is to use genetic analysis via manipulating gene activity spatiotemporally and then observe the functional consequences. Accordingly, the availability of genetic tools for such manipulations is critical to dissect the molecular and genetic mechanisms underlying the specificities and intricacies of an organism. Because of its powerful genetics as well as other unique features, including its short life cycle, invariant cell lineage, and well-annotated genome, *Caenorhabditis elegans* is an important genetic model organism for biological research and has contributed to new insights into many fundamental biological processes [1]. Development of innovative genetic tools that allow simultaneous gene disruption and transgene control will greatly facilitate scientific research to understand gene function.

Throughout the years, genetic tools for *C. elegans* researchers continue to be developed and improved [2]. Different approaches have been used to generate alleles for different *C. elegans* genes to understand their functions [3]. For example, forward genetic screens using the chemical EMS have isolated mutants that define many classic alleles [4,5]. Later, large-scale reverse genetic screens by the *C. elegans* Deletion Mutant Consortium, which use different mutagens to induce mutations followed by deletion detection, have also generated alleles for about 1/3 of the ~20,000 genes encoded in the *C. elegans* genome [6]. More recently, the CRISPR/Cas9 genome editing technology can be used to create mutations in virtually any *C. elegans* genes [7]. While these powerful tools can manipulate endogenous gene activity by creating mutations for functional genetic studies, the resultant alleles cannot be directly used for transgene control.

Similarly, tools for transgenesis and transgene control for *C. elegans* have been improving as well. Classic transgenesis in *C. elegans* is achieved by microinjection of DNA constructs with direct promoter-gene fusion to create multi-copy extrachromosomal arrays [8]. Later, transposon *Mos1*-mediated single-copy insertion, CRISPR/Cas9-mediated homologous recombination, integrase-mediated and recombinase-mediated cassette exchange (RMCE) have also been developed to generate single-copy transgenes [9–12]. For transgene control, several bipartite expression systems, including Q/QUAS, GAL4-UAS, Tet-Off (and Tet-On)/TetO, and LexA/LexO, have successfully been adopted for *C. elegans* in the past decade [11,13–15]. Each of these systems contains their own sets of driver strains and effector strains: in a driver strain, a heterologous transcription factor (e.g., cGAL for the GAL4-UAS system for *C. elegans* [14]) being expressed under a tissue- or cell-specific promoter; in an effector strain, the gene of interest is under the control of an enhancer sequence (e.g., UAS for the cGAL system) that will be specifically recognized by the transcription factor in the driver strain. Only in the cross progenies that contain both the driver and the effector will the gene of interest be expressed in specific cells. Because of the combinatorial feature, these transgene

technologies allow efficient generation of strains by crossing different driver and effector strains to understand gene function [2]. However, there is no genetic approach for *C. elegans* researchers to disrupt gene activity and simultaneously provide genetic access to cells that express the disrupted gene for controlling transgene expression; such a genetic approach will dramatically increase the efficiency of genetic studies to reveal the function and site of action of the gene of interest. In addition, there are no tools for reusing the same regulatory elements for transgene control by converting any existing driver to a new driver of a different bipartite expression system *in vivo* in *C. elegans.*

Gene trap was originally designed and applied as a random insertional mutagenesis technique to clone new genes and understand their functions [16,17]. Canonically, a gene trap vector contains a splice acceptor, a reporter (e.g., *LacZ* and *gfp*), and a transcriptional termination sequence. Insertion of the vector into an intron of a gene generally disrupts the gene activity because the added splice acceptor would presumably hijack the endogenous splicing to create a fusion mRNA with upstream exons of the target gene and the reporter. If the reporter has the same reading frame as the target gene, the reporter will be expressed under the control of the endogenous promoter, revealing the expression pattern of the target gene [16]. However, such a gene trap technique has not been adopted for *C. elegans* [18].

Here, we describe a cGAL-and-operon-based gene trap strategy that can be specifically targeted to presumably any gene in *C. elegans* using CRISPR/Cas9-mediated homologous recombination. The resulting gene trap lines are versatile for genetic analysis: they can 1) become loss-of-function alleles of target genes, 2) control expression of different transgenes for various experimental purposes, when combined with different UAS effector lines, and 3) be further engineered *in vivo* to be swapped with other bipartite expression systems through RMCE.

## Results

### Design of cGAL-based gene trap

We designed a cGAL gene trap cassette that can be inserted into a defined location in the genome through CRISPR/Cas9 genome editing and can be swapped with other constructs of interest through DNA recombinase Flippase (Flp)-mediated RMCE (Fig 1). Our design of the cGAL gene trap (cGAL GT) cassette includes a FRT (Flp Recognition Target) site, a synthetic splice acceptor [19], a short artificial exon, multiple stop codons in all three reading frames [7], an SL2 trans-splicing sequence from the *C. elegans gpd-2/gpd-3* operon [20], an optimized cGAL transcription factor with a *tbb-2* 3' UTR [21], a self-excising cassette (SEC) as a temporary transgenic marker [10], and a FRT3 site in the reverse direction. The cGAL gene trap components are modular and SapTrap compatible, enabling easy and efficient cloning of repair templates for CRISPR knock-in via SapTrap, a type of Golden Gate Cloning [19]. Each repair template also contains two homology arms (5' HA and 3' HA) that are specific for each CRIPSR-targeted insertion site in the *C. elegans* genome. Through CRISPR/Cas9-mediated homologous recombination followed by SEC removal via heat shock [10], a cGAL gene trap line is generated by inserting the cGAL gene trap cassette specifically into an intron of the target gene (Figs 1A and S1).

Due to the SL2 *trans*-splicing sequence, the resultant cGAL gene trap line functions like a bicistronic operon that would simultaneously have two outcomes (Fig 1A). First, alternative splicing to the added splice acceptor in the cGAL gene trap cassette would produce a truncated protein, in most cases, creating a loss-of-function allele of the target gene. Second, the bicistronic operon would express cGAL in the same cells where the target gene is endogenously expressed, creating a cGAL driver that can be combined with UAS effector lines to express virtually any effector gene in those cells.

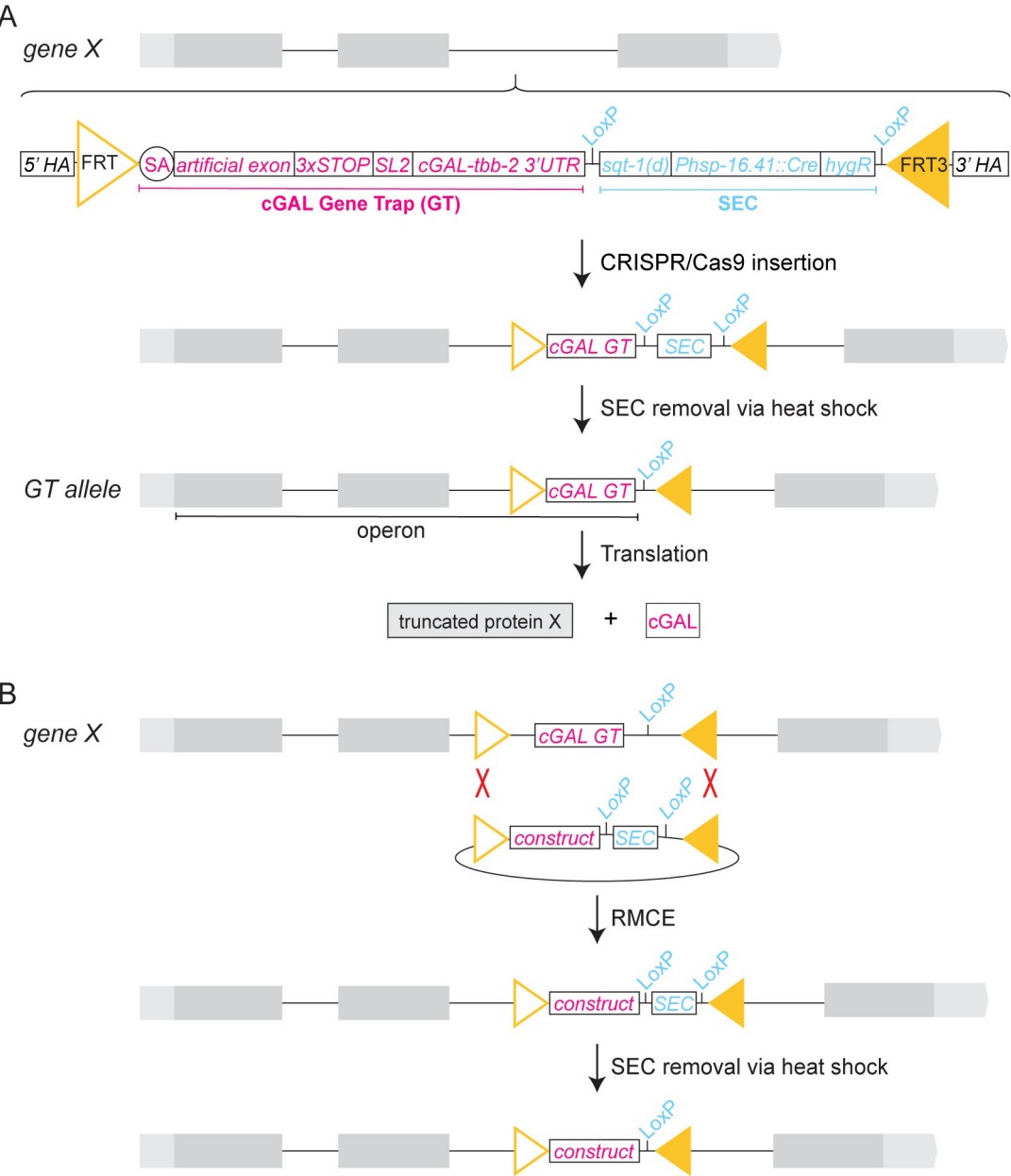

**Fig 1. Design of the cGAL gene trap strategy. (A)** Schematic of creating a cGAL gene trap (cGAL GT) allele via CRISPR/
Cas9-mediated homology-directed repair using two homology arms (5' and 3' HAs). The grey rectangles, black lines, and light
grey boxes represent exons, introns, and UTRs of the target gene *X*, respectively. The cGAL GT construct includes a splice
acceptor (SA), a short artificial exon, a sequence with stop codons in all three reading frames (3xSTOP), an SL2 *trans*-splicing
sequence from the *C. elegans gpd-2/gpd-3* operon, and cGAL [cGAL(DBD)-linker-QFAD] with a *tbb-2* 3' UTR. The self-
excision cassette (SEC), flanked by two LoxP sites of the same direction, contains a dominant Roller gene (*sqt-1(d)*), a heat
shock-induced Cre transgene (*Phsp-16.41::Cre*), and a hygromycin resistance gene (*hygR*). The SEC serves as a transgenic
marker to ensure the integration of the cGAL GT construct into the genome by identifying hygromycin-resistant Rollers. The
cassette with cGAL GT and SEC is flanked by an FRT site and a reverse FRT3 site, which allows further engineering of the
resultant gene trap line with RMCE. After the cGAL GT construct is integrated into an intron of the target gene *X*, homozygous
Roller animals are heat shocked to induce Cre expression in the germline to remove SEC, generating a final cGAL gene trap
allele. Due to the splice acceptor and the SL2 *trans*-splicing sequence in cassette, the resultant cGAL gene trap allele acts like a
bicistronic operon that would produce a truncated protein X and cGAL in the cells where the target gene *X* is endogenously
expressed. **(B)** Schematic of swapping cGAL GT with other DNA constructs *in vivo* via recombinase-mediated cassette
exchange (RMCE). The cGAL GT line acts as a landing site to swap cGAL GT with any DNA sequences flanked by an FRT site
and a reverse FRT3 site, including gene trap constructs with other bipartite expression systems (see Fig 4). With the target
RMCE construct and the DNA recombinase Flippase (Flp) in the germline, the cGAL GT can be reliably swapped with the

target construct through the flanking FRT and FRT3 sites. After successful RMCE, Roller animals are heat shocked to remove the transgenic marker SEC, generating non-Roller animals with the new transgene of the target construct.

In addition, because of the flanking FRT site and reverse FRT3 site, cGAL gene trap alleles can be further modified *in vivo* through Flp-mediated RMCE that has been recently adopted for *C. elegans* transgenesis [11]. Specifically, with Flp expressed in the germline, the cGAL gene trap cassette inserted in the *C. elegans* genome can be swapped with any other DNA construct flanked by the FRT and reverse FRT3 sites via RMCE (Fig 1B). Overall, the cGAL gene trap cassette allows for site-specific insertion to simultaneously generate loss-of-function alleles of target genes and provide genetic access to the cells expressing the target gene, and can be easily exchanged with other constructs of interest, demonstrating the versatility of the gene trap design.

## Generation of loss-of-function alleles for target genes

To determine if our cGAL gene trap design can presumably hijack the endogenous splicing of the target gene and thus generate a truncated gene product to produce a loss-of-function allele, we tested our design using the *aex-2* gene. It encodes a G protein-coupled receptor (GPCR) essential for the expulsion step to expel gut contents from the intestine during the rhythmic defecation motor program in *C. elegans*, and *aex-2(sa3)* loss-of-function mutants show an easily observable constipated phenotype ([22] and S2 Fig). We inserted the cGAL gene trap cassette in the third intron of the *aex-2* gene by CRISPR/Cas9-triggered homologous recombination to generate two independent alleles *tan74* and *tan75* (Fig 2A). Both alleles phenocopied the loss-of-function allele *aex-2(sa3)*: they showed strong defects in expulsion and were severely constipated (Figs 2B and S2). Furthermore, the *aex-2(tan75)* gene trap allele was recessive and failed to complement *aex-2(sa3)* (Fig 2B), confirming that the *tan75* allele is a loss-function allele of *aex-2*. Thus, these results show that insertion of our cGAL gene trap cassette in the intron of the target gene can create a loss-of-function allele.

## Genetic rescue of target genes

Next, we tested if we could use the cGAL gene trap alleles with corresponding UAS effector lines for genetic rescue experiments. The cGAL driver is presumably expressed under the promoter of the target gene, and thus, it should drive the expression of the UAS effector gene in the same cells where the target gene is expressed. We reasoned that the expulsion defects of *aex-2(tan75)* would be rescued if we combined the *aex-2* gene trap allele with a *UAS::aex-2(+)* effector transgene. We found that the expulsion defects are fully rescued by crossing *aex-2 (tan75)* mutants to *tanSi67*, a single-copy *11xUAS::aex-2(+) cDNA::linker::gfp::tbb-2 3' UTR* effector line (Fig 2C). As a negative control, the same *aex-2(+)* cDNA effector line alone cannot rescue the *aex-2(sa3)*, suggesting that successful rescue requires both the cGAL driver from the *aex-2(tan75)* gene trap line and the *tanSi67[11xUAS::aex-2(+)]* effector. We also showed that *aex-2(tan75)* could be rescued with a previously reported multiple-copy *UAS::aex-2(+)* effector expressed from an extrachromosomal array transgene (*syEx1444 [15xUAS::aex-2(+) cDNA]*, see [14]), though the rescue efficiency was more variable likely due to mosaicism of the array (S3 Fig). These results also show that cGAL gene trap alleles can be used with either single-copy or multi-copy UAS effector lines for genetic rescue experiments.

## Expression pattern of target genes

Additionally, we determined if cGAL gene trap lines can be used to study the expression pattern of the gene of interest. When crossing *aex-2(tan75)* with a single-copy UAS::fluorescent

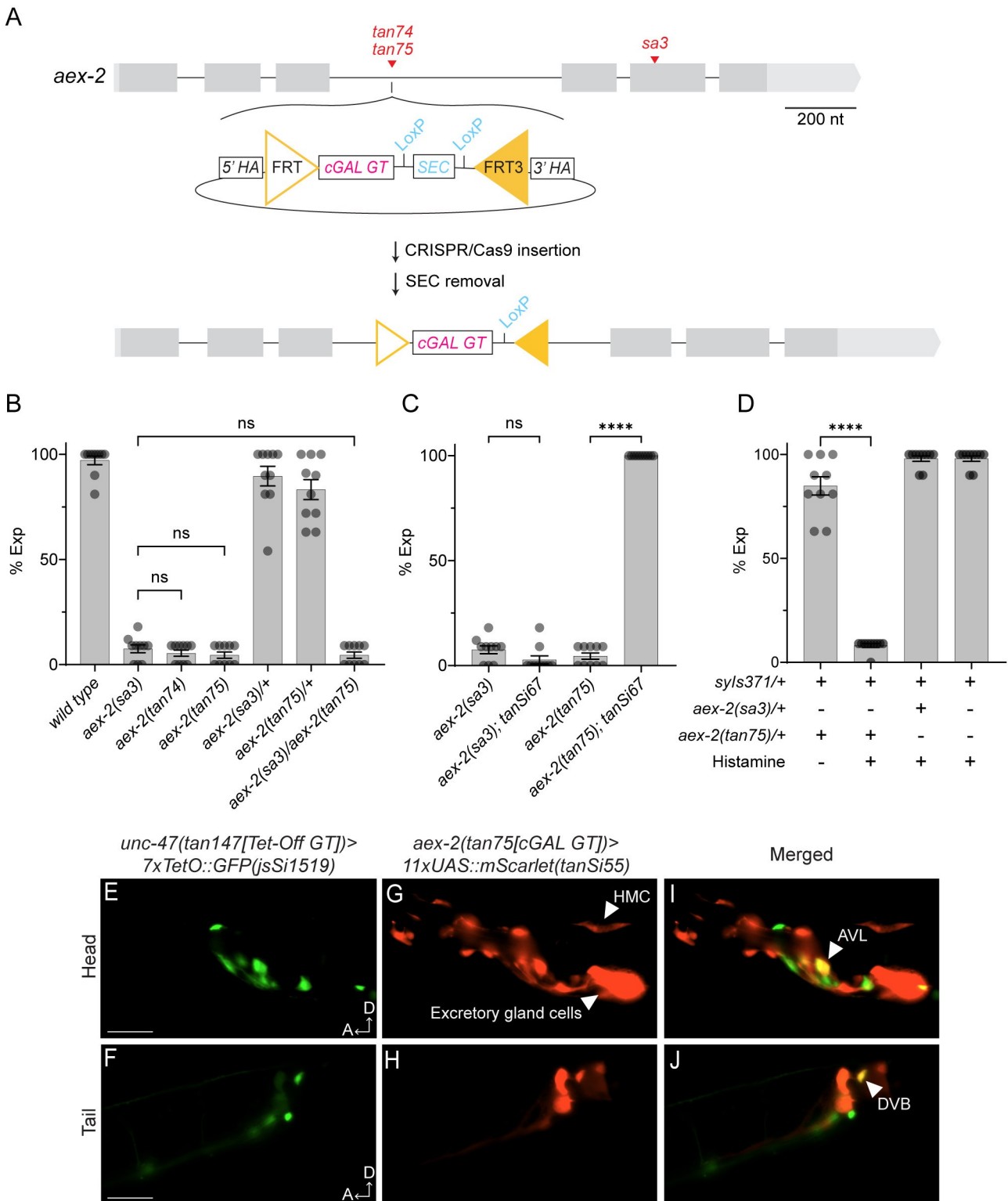

**Fig 2. Demonstration of the cGAL gene trap strategy in *aex-2*. (A)** Diagram showing the cGAL gene trap cassette was inserted into the third intron of *aex-2*, resulting in two independent alleles *aex-2(tan74)* and *aex-2(tan75)* post-SEC removal. *aex-2(sa3)* is a well-characterized missense allele, and its location is shown relative to the gene trap insertion location. The scale bar is 200 nucleotides. **(B-D)** Quantification of expulsion (Exp) defects of animals with indicated genotypes. Each dot represents a single animal (N = 10 worms). Mean ± SEM is shown. **(B)** *aex-2(tan75)* is a recessive loss-of-function allele and phenocopies *aex-2(sa3)*. ns, not significant. One-way ANOVA with post-hoc Tukey HSD. **(C)** The expulsion defects in *aex-2(tan75)* cGAL

gene trap mutants can be rescued with a single-copy *11xUAS::aex-2(+) cDNA::linker::gfp::tbb-2 3' UTR* effector (*tanSi67*). This rescue is not observed in *aex-2(sa3); tanSi67* animals. ns, not significant; ****, p<0.0001. One-way ANOVA with post-hoc Tukey HSD. (D) Silencing *aex-2* expressing neurons causes expulsion defects. The defects were observed only in the condition where double heterozygous animals with the cGAL gene trap allele *aex-2 (tan75)* and an integrated multi-copy *15xUAS::HisCl1* effector (*syIs371*) were treated with histamine, but not in the three negative controls shown. ****, p<0.0001. Paired Student's t-test. (E-F) Fluorescent images showing a subset of GABAergic neurons, including AVL in the head and DVB in the tail, in a marker strain with *unc-47(tan147[Tet-Off GT])>jsSi1519[7xTetO::GFP]*. (G-H) Fluorescent images showing cells expressing *aex-2* in the head and the tail regions of a strain with *aex-2(tan75[cGAL GT])>tanSi55[11xUAS::mScarlet]*. Notable cells include the head mesodermal cell (HMC), the excretory gland cell, the AVL neuron, several other head neurons (G), and the DVB neuron in the tail (H). (I-J) Co-localization of GABAergic neurons in (E-F) and *aex-2* expressing cells in (G-H), showing the expression of *aex-2* in AVL and DVB neurons, the previously reported sites of action of *aex-2* for the expulsion step. All fluorescent images shown follow the scale labeled in F. Scale bar is 50 μm. A, anterior; D, dorsal.

protein effector line, we detected fluorescence signals in head and tail neurons, enteric muscles, and head mesodermal cell (HMC) (Figs 2G, 2H and S4), which were not observed in animals with the UAS::fluorescent protein effector line alone (S5 Fig). This result is consistent with the expression pattern of *aex-2* previously reported using transcriptional and translational reporters [22]. In particular, when we crossed a red marker strain of the *aex-2* cGAL gene trap with an *11xUAS::mScarlet* effector to a green marker strain for GABAergic neurons (Fig 2E and 2F), we observed colocalization of the red and green fluorescence signals in two GABAergic neurons, AVL and DVB (Fig 2I and 2J), where *aex-2* functions to control the expulsion step [22,23].

In addition, we also found that *aex-2* is strongly expressed in the binucleate excretory gland cell (Figs 2G and S4), which has not been reported previously [22]. This is consistent with the presence of *aex-2* mRNA in the gland cell as reported by a recent single-cell transcriptional atlas of adult *C. elegans* [24], although the modest *aex-2* mRNA level detected in the excretory gland cells does not correlate with the high intensity of the fluorescence signal in the excretory gland cells from the *aex-2* cGAL gene trap. We further confirmed the expression of *aex-2* in the excretory gland cell (S6 Fig), using an endogenously tagged reporter allele *aex-2(syb4447 [aex-2::SL2::GFP::H2B])*. This finding suggests that there may be a biological role for *aex-2* within the excretory gland cell that has yet to be discovered.

The use of cGAL gene trap lines for expression pattern analysis is advantageous: it is much more robust than using a single-copy insertion of fluorescent proteins at the endogenous locus (S4 Fig), as the cGAL system allows for amplification of expression of the reporter in the UAS effector line. Thus, the cGAL gene trap lines can be used to reliably report the expression pattern of the target gene.

## Functional studies of target genes

Next, we tested if cGAL gene trap lines can be used to manipulate the neurons that express the target gene to reveal the function of these neurons. Activation of the two GABAergic neurons, AVL and DVB, by the AEX-2 signaling triggers the expulsion step [22,23]. Thus, we determined if we could use the *aex-2* cGAL gene trap line to inhibit *aex-2*-expressing neurons and tested if silencing these neurons phenocopies *aex-2* mutants. To this end, we crossed the cGAL gene trap line *aex-2(tan75)* with the transgene *syIs317*, a previously validated *15xUAS::HisCl1:: SL2::gfp* effector line [14], and quantified expulsion defects (Fig 2D). HisCl1 is a histamine-gated chloride channel from fruit flies, and it has been shown that *C. elegans* neurons expressing this heterologous channel can be silenced only in the presence of exogenous histamine [25]. Without supplementation of histamine, the *aex-2(tan75)/+; 15xUAS::HisCl1/+* double heterozygous animals showed normal expulsion during the defecation cycle. However, in the presence of 10 mM histamine, these animals were completely defective in expulsion (Fig 2D), similar to *aex-2* mutants. As negative controls, *aex-2(sa3)/+; 15xUAS::HisCl1/+* double heterozygous animals and *15xUAS::HisCl1/+* animals showed normal expulsion even in the presence

of 10 mM histamine (Fig 2D). These results suggest the *aex-2* cGAL gene trap line can be combined with an *UAS::HisCl1* line to silence *aex-2*-expressing neurons to cause expulsion defects, consistent with previous findings that *aex-2* activates AVL and DVB neurons to trigger the expulsion step [22,23]. Thus, cGAL gene trap lines can be used to manipulate neurons and cells when crossed with corresponding UAS effector lines.

### Robustness of cGAL-based gene traps

We tested if the cGAL gene trap strategy works robustly when the cassette is inserted into other *C. elegans* genomic loci. We selected two additional genes to test, *unc-47* and *inx-16*, both of which are important for the defecation motor program.

*unc-47* encodes a vesicular GABA transporter and is expressed in all GABAergic neurons. *unc-47* loss-of-function mutants are uncoordinated and severely defective in the expulsion step of the defecation motor program [26]. We used CRISPR/Cas9 to insert the cGAL gene trap cassette into the first intron of *unc-47* to generate a cGAL gene trap allele *tan93* (Fig 3A). The *tan93* allele phenocopied a classic *unc-47(n2409)* allele and was defective in the expulsion step (Fig 3C). Like the *aex-2(tan75)* cGAL gene trap line shown above, *unc-47(tan93)* can reveal the expression pattern of *unc-47* when crossed with the *UAS::gfp* effector line (*tanSi34*): the *unc-47(tan93)/+; UAS::gfp/+* double heterozygotes showed strong green fluorescence in GABAergic neurons (Fig 3E), consistent with previous results that used a traditional *unc-47* transcriptional reporter [26].

*inx-16* encodes a gap junction protein that localizes specifically to cell-cell contacts between intestinal cells, and loss-of-function *inx-16(ox144)* mutants are defective in the expulsion step [27]. We generated two cGAL gene trap alleles at distinct locations: one is within the first intron (*tan174*) and the other one lies in the second intron (*tan99*) (Fig 3B). We would like to note that in the *inx-16(tan174)* allele, the cGAL gene trap cassette was inserted only 25 base pairs downstream of an endogenous splice donor site. Nonetheless, the expulsion defects observed in both *tan174* and *tan99* alleles were comparable to the *inx-16(ox144)* allele (Fig 3D). Additionally, when the cGAL gene trap line *inx-16(tan99)* or *inx-16(tan174)* was crossed with the *UAS::gfp* effector line *tanSi34*, the GFP expression pattern showed *inx-16* is exclusively expressed in the intestine (Fig 3F and 3G), which agrees with previous findings [27]. Thus, the cGAL gene trap strategy is robust.

### Conversion to new gene traps with other bipartite expression systems by cassette exchange

We wanted a way to reuse the regulatory elements of the target genes revealed by the existing cGAL gene trap to create drivers with the corresponding transcription factors from other bipartite expression systems that have been shown to work in *C. elegans* as single-copy transgenes, including QF, QF2, LexA, Tet-Off and Tet-On [11]. Because the cGAL gene trap cassette is flanked by an FRT site and a reverse FRT3 site, each cGAL gene trap line can serve as a landing site for Flp-mediated RMCE (Figs 1B and 4A). As such, we tested if we could apply RMCE to directly convert existing cGAL gene trap lines into gene traps with other different transcription-based bipartite expression systems. Initially, we used the RMCE procedure in *C. elegans* via microinjection [28]. We first crossed the germline-expressing Flp transgene *bqSi711* [29] into cGAL gene trap lines to make double homozygotes. We injected a new RMCE donor construct with the SEC positive selection marker (e.g., Tet-Off GT) into the gonad of these animals and selected for homozygous transgenic worms with the SEC marker in the next generations. This method worked as expected with a similar RMCE efficiency as in previous studies [11,28].

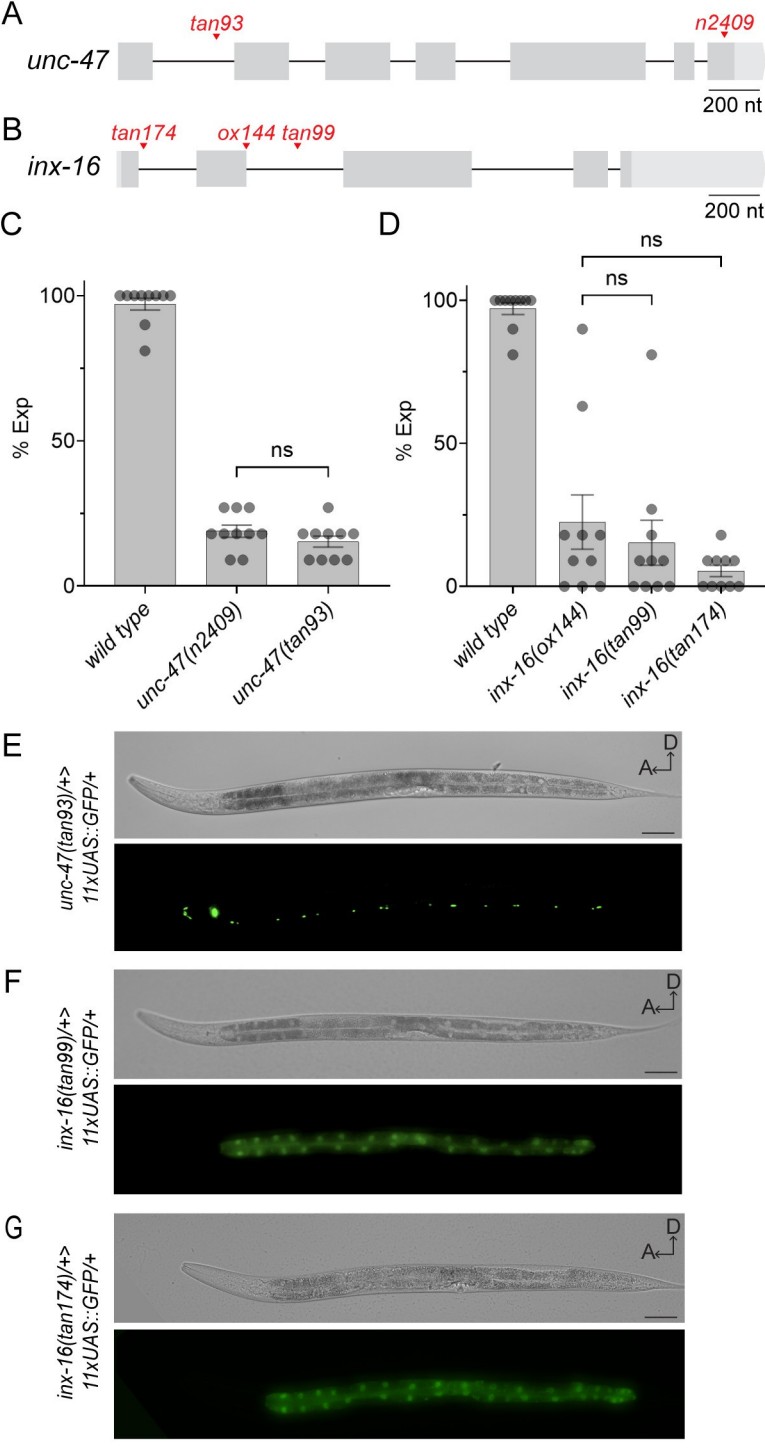

**Fig 3. Robustness of the cGAL gene trap strategy. (A-B)** Diagrams showing the cGAL gene trap lines for *unc-47* and *inx-16*. In *unc-47(tan93)*, the cGAL gene trap cassette was inserted into its first intron (A); in *inx-16(tan174)* and *inx-16(tan99)*, the cassette was inserted into its first and second introns, respectively (B). The locations of loss-of-function alleles *unc-47(n2409)* and *inx-16(ox144)* are shown relative to the gene trap insertion locations. The scale bar is 200 nucleotides. **(C-D)** Quantification of expulsion (Exp) defects of animals with indicated genotypes. Each dot represents a single animal (N = 10 worms). Mean ± SEM is shown. ns, not significant. One-way ANOVA with post-hoc Tukey HSD. The expulsion defects of *unc-47(tan93)* are similar to *unc-47(n2409)*, showing that *unc-47(tan93)* is a loss-of-function allele (C). The expulsion defects of *inx-16(tan99)* and *inx-16(tan174)* are similar to *inx-16(ox144)*, showing that both gene trap alleles are loss-of-function alleles (D). **(E-G)** DIC and fluorescent images showing the expression

patterns of *unc-47* and *inx-16* by crossing the corresponding gene trap lines with a single-copy *11xUAS::GFP* effector line (*tanSi34*). Double heterozygotes of *unc-47(tan93)/+; tanSi34/+* show green signal specifically in the GABAergic neurons (E). Double heterozygotes of *inx-16(tan99)/+; tanSi34/+* and *inx-16(tan174)/+; tanSi34/+* show green signal specifically in the intestine cells (F-G). Scale bar is 50 μm. A, anterior; D, dorsal.

However, each gene trap swap via RMCE requires a separate microinjection, which is tedious. Because DNA constructs injected into *C. elegans* can form extrachromosomal arrays that can be transmitted across generations [8], we sought to test if the RMCE step could be done by simple genetic crossing. To do so, we first created an extrachromosomal array with an RMCE donor construct and used it for genetic crossing to generate animals with the three components: the array, a cGAL gene trap allele, and the germline-expressing Flp transgene *bqSi711*. This strategy included both positive and negative selection markers to identify the animals with successful RMCE (S7 Fig). Indeed, we found that the efficiency of RMCE for gene trap swapping by genetic cross ranged from 10% to 100%, with an average of ~47% (36 out 76 cross plates, S1 Table). Thus, the efficiency of RMCE by genetic crossing is comparable, if not higher, than that of previously reported RMCE via microinjection [11,28].

Finally, we tested if new gene traps lines generated through RMCE function like the original cGAL gene trap lines. To do so, we compared the *unc-47(tan93[cGAL GT])* allele with other gene trap lines that were generated by swapping out cGAL gene trap in *unc-47(tan93)* via RMCE. Successful swaps to transcriptional factors from other bipartite expression systems in these new gene trap lines were validated using Oxford Nanopore linear amplicon sequencing of PCR products containing the section between the FRT site and the reverse FRT3 site as well as a few hundred base pairs of upstream and downstream flanking genomic DNA sequences. We found that all gene traps with other transcription factors produced a loss-of-function phenotype comparable to the cGAL gene trap mutants (Fig 4B). When combined with their corresponding GFP effector lines, these new gene traps also drive GFP expression in the GABAergic neurons (Fig 4C), largely similar to the *unc-47* cGAL gene trap (Fig 3E), suggesting these gene trap lines all reflect the endogenous expression pattern of *unc-47*. For reasons unknown, GFP signal in the GABAergic neurons in the head region of the *unc-47* LexA gene trap appeared to be much dimmer than the signal in those neurons along the ventral cord. The different effector strains we used for these new *unc-47* gene trap lines are from a previous study and they alone have minimal background expression, if any [11]. Thus, we demonstrate the feasibility and effectiveness of RMCE through genetic cross in *C. elegans*. We also show that we can use RMCE to effectively convert cGAL gene trap lines into other gene trap lines *in vivo*.

## Discussion

Here we describe the first cGAL-based gene trap strategy in *C. elegans*, which can disrupt the activity of the target gene and at the same time serve as cGAL drivers to provide genetic access to the cells where the targeted gene is expressed. When combined with different UAS effector lines, the cGAL gene trap lines can drive the expression of different transgenes for various experimental purposes, such as rescue of mutant phenotypes, analysis of gene expression patterns, and manipulation of neuronal activity. Our design also allows further engineering of cGAL gene trap alleles *in vivo* to efficiently convert them into gene traps with other bipartite expression systems via RMCE. We also further show that RMCE in *C. elegans* can be done simply by genetic crossing, in addition to the previously reported procedure through microinjection. The combinatorial feature of genetic cross enables efficient conversion of existing and future cGAL gene trap lines to other types of gene traps via RMCE, without the need to generate each individual gene trap line by CRISPR-mediated genome editing from scratch. This

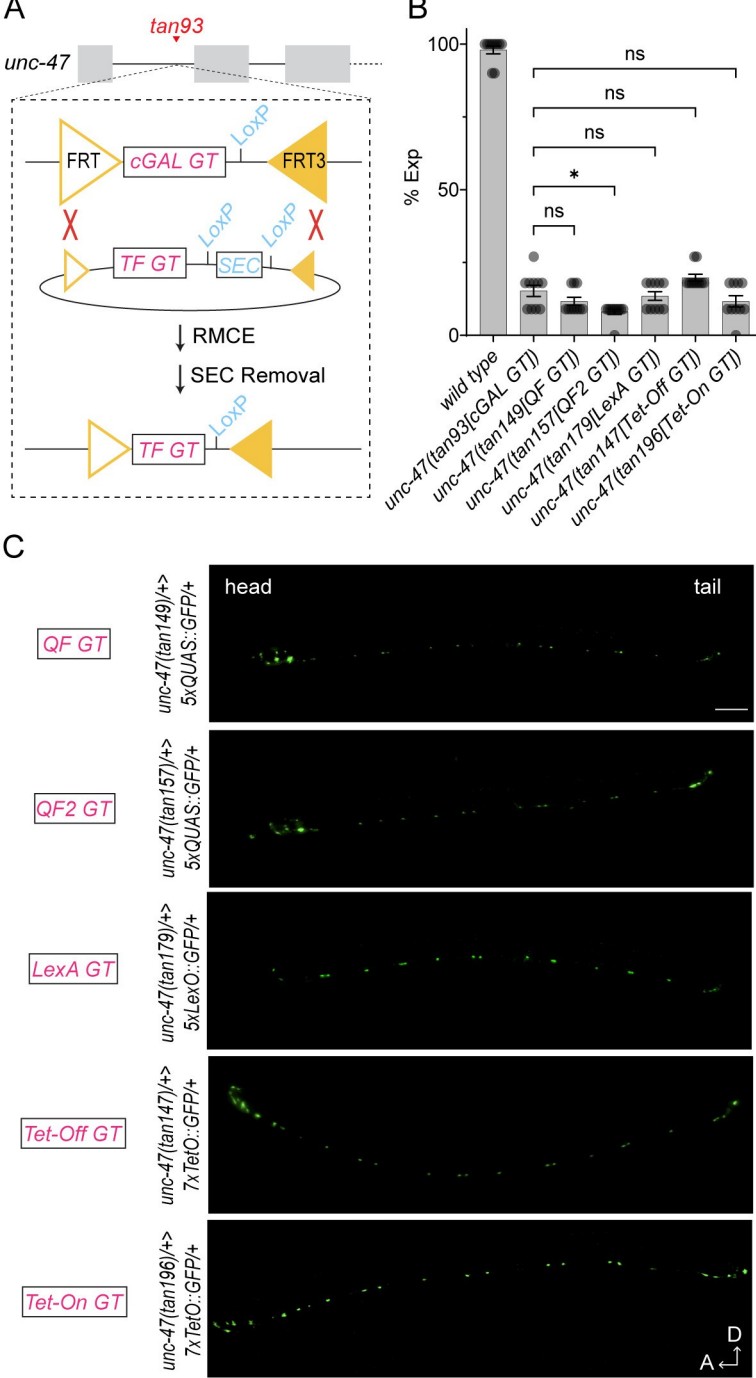

**Fig 4. Converting cGAL gene traps to new gene traps with different bipartite expression systems via RMCE. (A)**
Schematic showing generation of new gene trap lines with transcription factors (TFs) from different bipartite
expression systems by performing recombinase-mediated cassette exchange (RMCE) in existing cGAL gene trap lines
(*e.g.*, *unc-47(tan93[cGAL GT])*) shown here). With Flp recombinase expressed in the germline, RMCE will occur at the
FRT and reverse FRT3 sites, leading to the exchange of the cGAL gene trap with other gene trap constructs. The
RMCE process can be done through microinjection or genetic crossing. After successful recombination, the temporary
transgenic marker SEC is removed by heat shock to generate final strains with new gene trap lines. GT, gene trap; TF,
transcription factor; SEC, self-excising cassette. (**B**) Quantification of expulsion (Exp) defects of animals with indicated
genotypes. Each dot represents a single animal (N = 10 worms). Mean ± SEM is shown. All new *unc-47* gene trap lines
with transcription factors from different bipartite expression systems were generated by swapping out the cGAL gene
trap in the original *unc-47(tan93)* allele via RMCE, and they show expulsion defects similar to *unc-47(tan93)*. ns, not

significant; *, p<0.05. One-way ANOVA with post-hoc Tukey HSD. (**C**) Fluorescent images of GABAergic neurons in worms with indicated genotypes. All gene traps are inserted in the same locus in the *unc-47* gene as the *tan93* allele shown in (A). When crossed with their corresponding GFP effector strains, these gene trap lines with different bipartite expression systems display similar expression patterns in GABAergic neurons. Scale bar is 50 μm. A, anterior; D, dorsal.

makes high throughput *in vivo* conversion of gene traps in *C. elegans* feasible and lowers the energy barrier for other labs to use this versatile cGAL gene trap.

## Considerations for the insertion site of the cGAL gene trap cassette

In our study, we tested our cGAL gene trap design by inserting the cGAL gene trap cassette in introns of the target genes and our results showed that gene trap works robustly for all the genes tested. For example, both *inx-16(tan99)* and *inx-16(tan174)*, with the cGAL gene trap cassette inserted at different introns, showed the same mutant phenotype and expression pattern (Fig 3). We recommend considering the following factors when selecting an insertion site to generate a new gene trap for the gene of interest: 1) select an intron of your target gene with a specific guide RNA sequence (http://genome.sfu.ca/crispr/search.html, see [30]) to insert the gene trap cassette into the genome via CRISPR; 2) if the main goal is to use the gene trap as a "promoter trap" by only focusing on the enhancer and regulatory elements in the promoter, then choose an insertion site within an intron closer to the start of the gene; 3) if the main goal is to use the gene trap to capture as many regulatory elements within the introns of the gene in addition to the promoter, then choose an insertion site within an intron closer to the end of the gene; 4) if the main goal is to use gene trap as a loss-of-function mutant, then choose an insertion site upstream of the coding sequence for critical domains of the target gene. Furthermore, as the cGAL gene trap cassette contains an added splice acceptor followed by multiple stop codons in all three reading frames, in principle, the gene trap strategy should also work even if the cassette is inserted into exons of target genes.

## Advantages of the cGAL gene trap strategy

The current design of the cGAL gene trap insertion cassette is versatile. First, the cGAL gene trap can be specifically targeted to presumably any genes in the *C. elegans* genome by CRISPR/Cas9-mediated homologous recombination. Second, the cGAL gene trap cassette includes stop codons in all three reading frames followed by an SL2 *trans*-splicing sequence, which creates a synthetic operon. A functional cGAL driver will always be expressed under the control of the promoter of the target gene, regardless of its reading frame. In contrast, for gene traps that use direct protein fusion or the translational skip sequence T2A, such as the T2A-GAL4-based CRIMIC gene trap strategy in *Drosophila*, the target gene and the inserted reporters or transcription factors need to have the same reading frame and accordingly, three different insertion constructs are needed [16,31]. As such, our operon-based gene trap design is particularly useful and effective for obtaining functional gene trap lines in efforts to generate large-scale gene trap lines through random insertion. As operons have been identified in many eukaryotes [32], our operon-based cGAL trap can potentially be adopted in other eukaryotic model organisms. Third, the design enables directional swapping to generate different types of gene traps via Flp-mediated RMCE, due to the combination of FRT and reverse FRT3 sites flanking the cGAL gene trap. This is different from the bidirectional cassette exchange using integrase ΦC31-mediated recombination with *attP* and *attB* sites used in *Drosophila* [33,34]. In addition, the design is future-proof because better optimized or new bipartite expression systems can be easily adopted for gene traps by conveniently swapping with existing cGAL gene trap lines.

### Potential limitations of the cGAL gene trap strategy

There are some potential limitations of our cGAL gene trap strategy. First, although regulatory elements that control expression for most *C. elegans* genes lie within the first few kilobases upstream of the start codon [2], introns may also contain regulatory elements that control gene expression [35]. Because the inserted cGAL gene trap cassette contains a transcriptional terminator in the *tbb-2* 3' UTR, depending on the insertion sites—for example, those with relatively large introns of the targeted gene after the insertion site—the cGAL gene trap may not fully capture all the regulatory elements for the expression of the target gene. Second, the 3' UTRs used for the inserted bipartite expression systems, including *tbb-2* 3' UTR or *act-4* 3' UTR, are originally designed to allow robust expression of single-copy drivers across all tissues [11]. However, 3' UTRs can also be important for controlling gene expression, particularly in the germline [36]. Thus, there may be a slight difference between the expression pattern of the endogenous gene and the one revealed using the cGAL gene trap. Nonetheless, genetic rescue experiments by combining the cGAL gene trap line with the corresponding effector line with a wild-type copy of the targeted gene can help alleviate such potential limitations.

### Outlook for the cGAL gene trap strategy

Our cGAL-based gene trap strategy provides a powerful way to interrogate gene function, examine gene expression, and enable precise transgene control. The cGAL gene trap lines can easily be combined with growing numbers of available UAS effector lines, which includes more than 30 validated UAS effector lines for manipulating neuron activity, ablating cells, expressing various fluorescent reporters, and perturbing specific signaling transduction pathways [14,37,38]. New cGAL gene trap lines can be generated by inserting the cGAL gene trap cassette into specific genes using CRISPR/Cas9-mediated homologous recombination or random locations across the genome using the miniMos insertion system [10,39]. We envision that the gene trap strategy we reported here will be invaluable for the community and will greatly increase the speed and efficiency of genetic analysis in *C. elegans* to answer the overall question of how genes in the genome regulate specific biological processes.

## Materials and methods

### Strain maintenance

All *C. elegans* strains used in this study were maintained at room temperature on NGM plates seeded with *E. coli* strain OP50, as previously described [4]. A detailed list of strains used and generated can be found in S2 Table.

### Molecular biology

Plasmids were constructed using empty entry vectors and destination vectors for SapTrap [11,19]. Some DNA fragments were synthesized (Twist Bioscience, San Francisco, CA or Integrated DNA Technologies, Coralville, IA). Inserts cloned into entry vectors were generated by standard molecular cloning procedures with PCR and Gibson assembly (New England Biolabs, Ipswich, MA) and subsequently transformed into home-made competent cells DH5α. Repair templates for CRISPR knock in and plasmids for gene trap swapping via RMCE were generated by SapTrap [19], a Golden Gate cloning procedure. Positive clones were verified via Sanger sequencing (Functional Biosciences, Madison, WI) or Oxford Nanopore long read sequencing (Plasmidsaurus, Eugene, OR). Some plasmids are available at Addgene (www. addgene.org). A complete list of plasmids and oligos can be found in S3 and S4 Tables.

## Generation and verification of transgenes

The standard microinjection procedure for *C. elegans* was used to inject DNAs into the gonads of adult worms to create transgenic worms [8]. Specific details for various strategies to generate gene trap lines are described below. While the LoxP flanked SEC is used as the transgenic marker to create gene trap lines in our design (Fig 1A), SEC flanked by other Lox sites (such as Lox2272 or Lox511I, see [40]) can also be used. We would like to note that reading frame 1 of the 51-bp artificial exon in the cGAL gene trap cassette encodes a short peptide containing a FLAG tag (S1 Fig). Thus, the truncated endogenous proteins in some gene trap lines may be tagged with a FLAG tag at the C-terminus. All other gene traps essentially have the same design as the cGAL gene trap shown in Fig 1A, except the reporter/transcription factor and 3' UTR included. GFP gene trap uses a *tbb-2* 3' UTR. Tet-On/Tet-Off, LexA, and QF/QF2 gene traps all use an *act-4* 3' UTR, as these transcription factors have been shown to perform well when coupled with the *act-4* 3' UTR for their corresponding bipartite expression systems [11].

Single-copy transgenes inserted in the gene trap lines reported here were confirmed by sequencing. Briefly, for each strain, a 60 mm NGM plate containing almost starved worms was washed with M9 buffer and lysed with worm lysis buffer and Proteinase K (NEB, Ipswich, MA). Genomic DNA was extracted using the Zymo *Quick*-DNA Miniprep Kit (D3024). The purified genomic DNA was used as template to amplify the target sequence containing the insertion as well as flanking genomic DNA sequences via PCR using a high-fidelity polymerase (Phusion or Q5 from NEB). The PCR products were sequenced using Oxford Nanopore long read sequencing (Plasmidsaurus, Eugene, OR) to confirm the insertion of the transgene. The gene trap lines reported here were also confirmed by functional verification with corresponding *gfp* effector lines [11,40].

## CRISPR/Cas9 knock-in strategy for creating cGAL gene trap

To generate gene specific cGAL gene trap lines, a recently optimized CRISPR/Cas9 knock-in method via the self-excising cassette (SEC) selection [41] was used with slight modifications. First, to select the site for insertion for the target gene, a CRISPR guide RNA selection tool (http://genome.sfu.ca/crispr/search.html) designed for the *C. elegans* genome [30] was used to select a specific guide sequence followed by the PAM site (5'-NGG-3') for Cas9. The 20-bp guide RNA sequence was cloned into the sgRNA vector pMLS134 [19] via Gibson assembly of the SapI digested vector and a synthetic 60-mer single-stranded DNA oligo with the guide sequence at the center and two 20-bp long homology arms with the vector using NEBuilder® HiFi DNA Assembly Master Mix (New England Biolabs, Ipswich, MA). Second, to generate a repair template for knock-in experiments, ~300–600 bp homology arms (HAs) of the *C. elegans* genome from the 5' and 3' ends of the Cas9 cut site selected were PCR amplified from N2 genomic DNA. The 5' HA and 3' HA PCR products were cloned into the entry vectors for Sap-Trap (slot 3 and slot 8, see S3 Table) via Gibson assembly. The repair template with 5' HA, FRT-splicing acceptor-artificial exon-3xSTOP-SL2, cGAL-*tbb-2* 3'UTR, SEC, reverse FRT3, and 3' HA was cloned in the destination vector pMLS257 via SapTrap assembly ([19] and S3 Table). Third, a typical injection solution was made as follows: 2.5 ng/μL of pCFJ90 (P*myo-2*::*mCherry*::*unc-54* 3'UTR, see [9]), 5.0 ng/μL of pCFJ104 (P*myo-3*::*mCherry*::*unc-54* 3' UTR), 5 ng/μL of the sgRNA plasmid, 25 ng/μL of the repair template, and 5 ng/μL of pDD121 (a germline-expressing Cas9 plasmid, see [10]), with all diluted in nuclease-free water (NEB, Ipswich, MA). Next, the solution was injected into ~40 young adult N2 (Bristol strain) worms ($P_0$) per cGAL gene trap knock-in, with 2 $P_0$ per plate, and the injected worms were placed at 25°C. After 3 days, 500 μL of 5 mg/mL hygromycin (Gold Biotechnology Inc., St. Louis, MO) was added to each plate for a final concentration of 0.25 mg/mL, and the plates were placed back at

25˚C for an additional 3–7 days. Candidate knock-ins were Rollers (Rol), lacked RFP extra-chromosomal arrays, and survived hygromycin selection. These were singled out on plates lacking hygromycin until they were homozygous for Rol. The SEC was excised by heat-shocking eggs and L1/L2 larvae at 30˚C for 18–20 hours [11]. After 3–5 days, non-Roller progenies were picked and singled out to establish non-Roller lines. Note that only one line from each $P_0$ plate was saved as an independent line.

## RMCE via microinjection for gene trap swapping

cGAL gene trap lines were crossed into the strain BN711 (*bqSi711[mex-5p::Flp::SL2::mNG + unc-119(+)] IV*), which expresses the DNA recombinase Flippase (Flp) and the green fluorescent protein mNeonGreen (mNG) in the germline, until they were homozygous for the cGAL gene trap allele (a landing site for RMCE) and *bqSi711*. The RMCE protocol was similar to the original RMCE protocol [11,28] but with a slightly different selection method. Briefly, DNA solutions with a RMCE gene trap plasmid of interest (gene trap of interest to swap in + SEC cloned in the vector pHW1133, see S3 Table) were injected at a total concentration of 50 ng/µL into 10 young adult worms ($P_0$). Injected worms were placed at 25˚C, with 2 $P_0$ per plate. After 3 days, 400 µL of 5 mg/mL hygromycin was added to each plate for a final concentration of 0.2 mg/mL, and the plates were placed back at 25˚C for an additional 3 days. Candidate RMCE knock-ins survived hygromycin selection and were Rol, and they were singled out on plates lacking hygromycin until they were homozygous for Rol. In general, we crossed out the *bqSi711* transgene, before characterizing the gene trap lines obtained from RMCE via microinjection. The SEC in the outcrossed lines without *bqSi711* was excised by heat-shocking eggs and L1/L2 larvae at 30˚C for 18–20 hours. After 3–5 days, non-Roller progenies were picked or singled out to establish non-Roller lines.

## RMCE via genetic cross for gene trap swapping

First, we generated a transgenic line with a "swapping" extrachromosomal array that contains the target RMCE gene trap plasmid and negative selection markers (HisCl1 expressing plasmids, see S3 Table). A typical injection solution for the array is 10 ng/µL of pHW1385 (P*myo-2::HisCl1::SL2::mScarlet-I-C1::tbb-2 3' UTR*), 10 ng/µL of pHW1391 (*Psnt-1::HisCl1::SL2::mScarlet-I-C1::tbb-2 3' UTR*), 50 ng/µL of the gene trap of interest with SEC, and 30 ng/µL of DNA filler pBlueScript KS(+) plasmid in nuclease-free water. For detailed cross schemes and the efficiency of RMCE via genetic crossing, refer to S7 Fig and S1 Table. Briefly, males containing both the cGAL gene trap allele and germline-expressing Flp transgene *bqSi711* were crossed into hermaphrodites with the "swapping" extrachromosomal array, and these crosses were placed at room temperature. For each gene trap of interest to swap in, 5 cross plates were set up to ensure 2–3 successful RMCE knock-ins. After 3 days of crossing (day 3 of the cross), each cross plate would have 400 µL of 5 mg/mL hygromycin added for a final concentration of 0.2 mg/mL. This selects Rol animals, whether cross progenies or self-reproduced progenies. After another 3 days (day 6 of the cross), 500 µL of 500 mM histamine was added to each plate at a final concentration of 25 mM [42] to select for successful knock-ins. Extrachromosomal arrays that survived would be unable to pump or move due to the P*myo-2::HisCl1::SL2::mScarlet-I-C1* with *Psnt-1::HisCl1::SL2::mScarlet-I-C1* or *Pmyo-3::HisCl1::SL2::mScarlet-I-C1* constructs included in the arrays. We recommend *Psnt-1::HisCl1::SL2::mScarlet-I-C1* over *Pmyo-3::HisCl1::SL2::mScarlet-I-C1* as it is more effective in blocking the movement of animals in the presence of histamine. Successful knock-ins consisted of rolling and pumping progenies with germline mNeonGreen (due to the *bqSi711* transgene) and no RFP signal that survived hygromycin and histamine selection. These progenies were singled out onto NGM plates seeded

with OP50 that did not contain hygromycin or histamine; after self-crossing, the plates were screened to ensure that worms were heterozygous for the RMCE knock-in and the *bqSi711* transgene. Then, worms would be singled out until they were homozygous Rollers and no longer contained the *bqSi711* transgene (no longer producing mNeonGreen in the germline). The SEC was excised by heat-shocking eggs and L1/L2 larvae at 30°C for 18–20 hours. After 3 days, non-Roller progenies were picked or singled out to establish non-Roller lines.

General recommendations for successful RMCE via genetic cross are as follows. When generating extrachromosomal arrays for RMCE knock-ins, we recommend choosing an extrachromosomal array line that has a reasonably good transmission rate. We also recommend testing the line by adding histamine onto a maintained NGM plate with adequate OP50 and observing if the Rol worms are paralyzed and not pumping shortly after the addition of histamine. In our hand, the inclusion of both *Pmyo-2::HisCl1::SL2::mScarlet-I-C1::tbb-2 3' UTR* and *Psnt-1::HisCl1::SL2::mScarlet-I-C1::tbb-2 3' UTR* in the array serves as a highly effective negative selection marker for arrays in the presence of histamine. Additionally, it is important that the cross plate is not starved and contains the food OP50. If the plate is nearly starving, then we recommend adding OP50 from an already-seeded NGM plate. We have also found that during the crossing process, it is normal to have extrachromosomal array worms that contain an RFP marker but do not Rol (likely due to mosaicism). It is also normal to find Rol male progenies or Rol hermaphrodite progenies that do not express germline Flp with successful RMCE knock-ins; we advise to not choose those worms when singling out successful RMCE knock-ins. We singled out those with germline-expressing Flp to ensure that the RMCE reaction is complete, and the obtained line will most likely only have a single-copy insertion of the swapping construct.

## Allele naming system for gene trap lines

As the targeted gene was being modified, all gene trap alleles generated from CRISPR/Cas9 knock-ins and RMCE swapping were designated a new allele name (*tan#* for our lab*)*. According to *C. elegans* nomenclature rules, RMCE insertions should be referred to as *tan#[parent strain]tan#[allele post-RMCE insertion]* but in the main text they were only referred to as *tan# [allele post-RMCE insertion]*. For example, *unc-47(tan157)* is technically *unc-47(tan93tan157)*, as the *tan93[FRT-cGAL GT-LoxP-rFRT3]* allele is the original *unc-47* cGAL gene trap allele that served as a landing site for RMCE to generate the new allele *tan157[FRT-QF2 GT-LoxP-rFRT3]*.

## Fluorescence imaging

Fluorescence imaging of L4 worms expressing GFP or mScarlet-I-C1 was performed in double heterozygous (gene trap/+; effector/+) animals except for the co-localization images, which were taken in double homozygous animals. L4 animals were mounted on a 2% agarose pad on a glass slide for imaging. First, worms were immobilized by transferring them to 20 μL of 30 mg/mL 2,3-Butanedione monoxime (BDM) solution (Sigma-Aldrich, St. Louis, MO) on a cover slide. When most worms were immobilized (~5–10 mins), a freshly prepared 2% agarose pad on glass slide was inverted onto the drop containing the immobilized worms, creating the imaging slide. All images were oriented with the anterior side of the worm to the left and the ventral side at the bottom. Imaging was performed using a 20X/0.75, WD 1 dry objective or 40X/1.3, ∞/0.17, WD 0.24 oil objective in a Nikon Ti2-E Eclipse inverted microscope connected to a Prime 95B (25mm) sCMOS camera. Green and red fluorescent images were taken with standard GFP and mCherry filter cubes, respectively, using a SPECTRA X LED light engine (Lumencor, Beaverton, OR). Representative images presented in the manuscript are

maximum intensity projections derived from Z-stack images of the sample worms, created using NIS-Elements Advanced Research software (Nikon). To induce GFP effector expression for the Tet-On gene trap experiment, 400 μL of 1 ng/μL doxycycline (Gold Biotechnology Inc., St. Louis, MO) was added to cross plates for a final concentration of 0.04 ng/μL and the worms were imaged after 24 hours.

### Behavioral assay

Defecation was assayed as previously described [23]. Briefly, 1-day-old adults were individually placed on fresh NGM plates seeded with OP50 and allowed to crawl for 5–10 minutes before the assay. Each animal was scored for 11 consecutive defecation cycles, and 10 animals were assayed for each genotype. Posterior body muscle contraction (pBoc) and expulsion (Exp) for each cycle were recorded using an EthoTimer program (Thomas lab website: http://depts. washington.edu/jtlab/software/otherSoftware.html). % Exp for each animal was calculated as the ratio of Exp to pBoc expressed as a percentage. One-way ANOVA was used to determine statistical significance unless otherwise noted. For the assay involving histamine, after scoring the defecation assay on regular NGM plates seeded with OP50, the animals were transferred to NGM plates containing 10 mM histamine seeded with OP50. After 1.5 hours of exposure to histamine, the animals were scored for another 11 consecutive cycles. A paired *t*-test was used to determine statistical significance between worms with and without histamine exposure.

### Blue food dye imaging for constipated phenotype

Briefly, 10 μL of M9 was added to a 1.5 mL Eppendorf tube, and OP50 from a seeded NGM plate was swirled into the M9 buffer. Then, about 50–75 adult worms were added to the M9 + OP50 mixture. After adding the worms, 30 μL of 20% (w/v) blue food dye stock (FD&C Blue #1 powder diluted in ddH$_2$O) was added to the tube. The worms + M9 + OP50 + blue food dye mixture was incubated in a shaking incubator for 30 minutes at 250 rpm, room temperature. After 30 minutes, the 40 μL mixture was added to a new NGM plate. Animals crawled around for 5–10 minutes before they were picked onto the center of a new seeded NGM plate. The area containing the worms was chunked and placed on a coverslip, and the live worms were imaged with a camera (Zeiss Axiocam 208 color) on an inverted injecting microscope (Zeiss Axio Vert. A1) using a 10x air objective.

### Statistical analysis

All statistical analyses were conducted using the software Prism 10 (GraphPad, Boston, MA). Raw data for all behavioral assays can be found in S1 Data.

## Supporting information

**S1 Fig. Annotated DNA sequence of the inserted cGAL gene trap cassette after SEC removal.** The "GCG" and "ACG" at the ends are the adaptor sequences from SapTrap cloning. SEC removal leaves a single LoxP site (blue highlight). >>, forward strand; <<, reverse strand.
(PDF)

**S2 Fig. Constipated phenotype of *aex-2* loss-of-function mutants.** Intestinal lumen was visualized by feeding adult worms with bacteria OP50 in blue food dye. For wild-type animals, the intestinal lumen of wild-type worms is thin; for *aex-2(sa3)* and *aex-2(tan75[cGAL GT])* mutants, the intestinal lumen is severely distended, particularly in the anterior and posterior sections of the intestine. Black arrows show an increased distention of the intestinal lumen of

*aex-2* mutants compared to wild-type worms. Scale bar is 50 μm. A, anterior; D, dorsal.
(TIF)

**S3 Fig. Genetic rescue of *aex-2(tan75)* using a multi-copy extrachromosomal array (*15xUAS::aex-2(+))* effector.** The expulsion defects in *aex-2(tan75[cGAL GT])* mutants can also be rescued with a multi-copy *15xUAS::aex-2(+)* extrachromosomal array effector (*syEx1444*). This rescue is not observed in *aex-2(sa3); syEx1444* animals. Each dot represents a single worm (N = 10 worms). Mean ± SEM is shown. ns, not significant; ****, p<0.0001. One-way ANOVA with post-hoc Tukey HSD.
(TIF)

**S4 Fig. Robustness of GFP expression from *aex-2* cGAL gene trap driver with an *11xUAS:: GFP* effector compared to *aex-2* GFP gene trap.** ZZZ538 is *tanSi34[11xUAS::GFP]; aex-2 (tan75[cGAL GT])* double homozygote, and ZZZ646 is *aex-2(tan102[GFP GT])*. Both cGAL in the *tan75* allele and GFP in the *tan102* allele were single-copy transgenes inserted at the same location in the third intron of *aex-2*. DIC images of the head region of both strains are shown (top panel). When digitally optimizing the look-up table to observe GFP expression in ZZZ538, no GFP expression is observed in ZZZ646 animals (middle panel). Alternatively, when optimizing the look-up table to observe dim GFP expression in ZZZ646, the GFP expression in ZZZ538 becomes overexposed (bottom panel). Arrowheads indicate the excretory gland cell. Scale bar is 50 μm. A, anterior; D, dorsal.
(TIF)

**S5 Fig. The UAS::GFP effector alone does not show GFP expression.** DIC and fluorescent images of the single-copy GFP effector strain ZZZ251 (*tanSi34[11xUAS::GFP]*). Scale bar is 50 μm. A, anterior; D, dorsal.
(TIF)

**S6 Fig. Expression of *aex-2* in the binucleate excretory gland cell.** Merged image of DIC and fluorescent images of the strain PHX4447, a transcriptional reporter strain of *aex-2*, in which a trans-spliced sequence encoding a nuclear localized GFP fusion (SL2::GFP::H2B) is inserted right after the endogenous *aex-2* locus. Very weak GFP fluorescence is observed in the two nuclei of the binucleate excretory gland cell, as indicated by two white arrows. Scale bar is 50 μm. A, anterior; D, dorsal.
(TIF)

**S7 Fig. Different genetic cross schemes for gene trap swapping by RMCE.** The most effective cross scheme is scheme 1, in which the males are double homozygous for the cGAL gene trap (cGAL GT) and the germline-expressing Flp transgene (*bqSi711*). Hygromycin serves as a positive selection marker for both the successful RMCE single-copy insertion as well as the array with the RMCE donor constructs; histamine serves as a negative selection marker for the array. Refer to see Materials and Methods for more details.
(PDF)

**S1 Table. Efficiency of gene trap swapping by RMCE via genetic crossing.**
(PDF)

**S2 Table. Strains used in this study.**
(XLSX)

**S3 Table. Plasmids used in this study.**
(XLSX)

**S4 Table. Oligos used in this study.**
(XLSX)

**S1 Data. Raw data for all behavioral assays.**
(XLSX)

## Acknowledgments

We thank D. Ehrlich, J. Nance, and the members of the Wang lab for comments on the manuscript. We thank M. Nonet (WUSTL) for sharing plasmids and strains. Some strains were provided by the CGC, which is funded by NIH Office of Research Infrastructure Programs (P40 OD010440). We thank WormBase for providing information about genome sequence and annotations.

## Author Contributions

**Conceptualization:** Haania Khan, Han Wang.

**Data curation:** Haania Khan, Xinyu Huang, Vishnu Raj.

**Formal analysis:** Haania Khan, Xinyu Huang, Vishnu Raj, Han Wang.

**Funding acquisition:** Han Wang.

**Investigation:** Haania Khan, Xinyu Huang, Vishnu Raj, Han Wang.

**Methodology:** Haania Khan, Xinyu Huang, Vishnu Raj, Han Wang.

**Project administration:** Haania Khan, Han Wang.

**Resources:** Haania Khan, Xinyu Huang, Han Wang.

**Supervision:** Haania Khan, Han Wang.

**Validation:** Haania Khan, Vishnu Raj, Han Wang.

**Visualization:** Haania Khan, Xinyu Huang, Han Wang.

**Writing – original draft:** Haania Khan, Han Wang.

**Writing – review & editing:** Haania Khan, Xinyu Huang, Vishnu Raj, Han Wang.

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
