## [Decision Letter · Decision Letter 0]

9 Oct 2024

Dear Dr Wang,

Thank you very much for submitting your Methods entitled 'A versatile site-directed gene trap strategy to manipulate targeted gene activity and control gene expression in C. elegans' to PLOS Genetics.

The manuscript was fully evaluated at the editorial level and by independent peer reviewers. The reviewers appreciated the attention to an important topic but identified some concerns that we ask you address in a revised manuscript.

We therefore ask you to modify the manuscript according to the review recommendations. Your revisions should address the specific points made by each reviewer.

To resubmit, log into your Editorial Manager account and select the option 'Revise Submission' in the 'Submissions Needing Revision' folder.

Yours sincerely,

Daniel A. Starr

Academic Editor

PLOS Genetics

Quanjiang Ji

Section Editor

PLOS Genetics

Thank you for sharing this study with PLoS Genetics. As you can see, all three reviewers see signficnat value for these tools to be used broadly by the C elegans community. All three also contain helpful comments in their reviews.

Please address all the reviewer comments. Then we would be very interested in seeing a revised manuscript.

Reviewer's Responses to Questions

**Comments to the Authors:**

Reviewer #1: Upload as an attachment

Reviewer #2: In the manuscript "A versatile site-directed gene trap strategy to manipulate targeted gene activity and control gene expression in C. elegans") the authors Khan et al. describe the development and testing of the first gene trap cassette in C. elegans. The gene trap cassette is elegant and versatile in its design and can simultaneously be used to generate null alleles and drive effector genes (via the cGal system). Furthermore, the cassette can easily be exchanged by RMCE for other driver systems. The authors convincingly demonstrate null phenotypes of three genes (aex-2, unc-47, and inx-16) and accurate capture of gene expression patterns using fluorophores. In addition, the authors demonstrate the significant utility of amplifying expression signals using a secondary single-copy 11x UAS::gfp transgene. Finally, the authors demonstrate efficient RMCE with a large number of transcription factor amplification systems (e.g., LexA, TetOn, TetOff) by injection and by a genetic crossing scheme.

Overall, the data shown are comprehensive and convincing. The manuscript is well-written with clear figures and is easy to comprehend, with extensive methods and protocol sections that will help other laboratories use the technique. An efficient gene trap method has been missing in the C. elegans toolbox and should be of significant utility to many laboratories in the field. I predict that the method will become widely used in the community.

One general comment about the gene trap design that the authors do not mention. As described, the splice acceptor and three-frame stop codons should truncate the protein. However, the downstream gpd-2/3 operon would be expected to do the same. The operon sequence leads to transcriptional termination and re-initiation. I do not know of any examples of operon sequences being "ignored," for example, for alternative splicing, so in effect, the authors have a "double termination" system. This may be worth including in the text, especially as the authors discuss the insertion site of the cassette relative to an endogenous splice site. It should, in principle, not matter given their design - including insertions into exons.

Suggested changes

I would mainly ask the authors to clarify and make the following additions to the text:

The methods mention validation of the insertion sites by PCR (and even a short discussion of using Oxford Nanopore to validate). There is no mention of the validation in the text. Please include a brief sentence in the results about whether all insertions were validated. Did the authors ever observe off-target insertions?

Specific comments:

Lines 206 - 213: Could the authors comment more exhaustively on the expression in the gland cells? There is a slight discrepancy with previously characterized expression patterns (although expression is consistent with more recent single-cell data but then the expression is higher than expected). Do the authors think this has to do with amplification of the fluorescence signal using the 11x UAS::mScarlet construct? For example, the minimal promoter used in the 11x UAS::mScarlet construct? Or could this be caused the 3' UTR regulation, either of the tbb-2 3' UTR in the cGal GT construct or the 11x UAS::mScarlet construct? Understanding how accurately the expression patterns reflect the endogenous expression is important for the utility of using the gene-trap cassette for further experiments.

Lines 217 - 218: The signal amplification from the 11x UAS::GFP construct is an important consideration. Could the authors comment and/or characterize the following:

A. Approximately how much amplification do they observe? They only mention different look-up tables, but it would be useful to have a "solid number" for how much signal amplification is observed, which should be easily inferred from the images and the microscopy settings on the already included images.

B. A second question concerns how many cells they see expression in. From the images in Fig. S4, it is difficult for the reader to see if the patterns are "similar" but just have different intensities. This is, of course, a difficult question to answer, but it would be useful for the reader to know how many cells are expected to express a-2 based on single-cell data and whether the patterns are consistent with this.

Lines 329 - 331: I agree that the gene trap cassette is versatile. However, could the authors comment on if and how the gene trap would be used in genes with lethal or sterile phenotypes? Are there ways to maintain the gene trap insertion for alleles that need to be heterozygous? The authors use the self-excising cassette (SEC), which contains a positive selection marker (HygroR). Using the SEC is convenient but would appear to be limiting for genes with a severe loss-of-function phenotype because the positive selection is removed before the genetically modified animals are "used" in experiments. Unless I am missing an important point, I would suggest including this in the list of limitations that comes immediately after in the discussion.

Lines 351 - 357: I am not sure I understand the logic of the discussion of the location of the enhancers relative to where the gene trap is inserted. If an enhancer is located in an intron downstream of the gene trap insertion (or further away), the enhancer could presumably still function to regulate transcription at the TSS. It is true that any perturbation of the genomic locus (including a "normal" N or C terminal gfp tag) can potentially disrupt transcriptional regulation, but that is difficult to predict based on the location. Perhaps the authors could elaborate on what exactly they mean.

Minor suggestions

p27. Figure 2, panel C. It would be useful to indicate the nature of the "tanSi67" transgene in the figure, even though the legend explains why these experiments should be rescued.

Line 84: Please add the allele name of the single-copy insertion (tanSi67)

Reviewer #3: In this work by Khan et al., the authors developed a versatile and neat gene trap system in the widely used genetic model organism C. elegans. In the C. elegans community, a method that allows simultaneous disruption of a target gene function and the expression of transgenes in the same cells where the target gene is disrupted is currently lacking. To address this key gap, the authors developed the cGAL-based gene trap strategy that can target any gene of interest using CRISPR/Cas9-mediated homologous recombination. Using an operon-based design, the cGAL gene trap cassette contains stop codons in all three reading frames to create loss-of-function allele of the targeted gene, and at the same time contains coding sequences for cGAL that would drive the expression of transgenes preceded by UAS in the same cell. This feature will allow for the analysis of expression patterns using UAS::gfp as well as the expression of transgenes to examine their ability of rescue mutant phenotypes. Importantly, the presence of Flp Rcognition Target sites flanking the gene trap cassette promises the ability to convert them into gene traps with other bipartite expression systems using recombinase-mediated cassette exchange. The authors convincingly presented data to demonstrate the efficacy and robustness of this system, and gained new biological insights, including the expression of aex-2 in the excretory gland cells. Overall this work is very well written and presented an innovative experimental system that could be of particular interest to the C. elegans community, with potentially broad impact to various biological problems using eukaryotic model organisms. Specific comments/points are as follows:

(1) It would be very helpful to the C. elegans field and any labs who would like to use this system if the authors can elaborate on the rationale behind the choice of introns or positions within introns where the cassette was inserted. In particular, it would be useful to know if the authors have tested different introns/locations and if so whether the results are comparable. It is noted that in Figure 3, both tan174 and tan99 were examined in terms of the inx-16 loss-of-function phenotypes, however, it is unknown whether the cGAL driven expression of UAS::GFP are similar (only one of them is shown in Fig. 3F). Answers to this question will be important to understand the extent of variability in the expression of cGAL, considering the lack of clarity in the locations of regulatory elements for the targeted genes in C. elegans (despite the authors’ mentioning in Line 178 – 179 that “The cGAL driver is presumably expressed under the promoter of the targe gene”).

(2) In Figure 4C, the expression pattern of most of the bipartite systems are similar to the GFP pattern in Fig. 3E, except LexA GT, where the prominent GFP signals in head GABAergic neurons are missing. Is the image representative? If so, the authors should provide interpretation or comments and consider modifying the text about all of them being the same as Fig. 3E.

(3) in Figure 2D, one more negative control where only syIs371/+ and Histamine are present would be required. More than one variable is changed between current groups.

Minor points:

(1) For all bipartite-driven expression pattern data, negative control lacking the driver should be shown (e.g. Fig. 2 GH, Fig. 3EF, Fig. 4C).

(2) in Figure S4, the choice of words “GFP (optimized for..)” is misleading. One option is to say “GFP intensity scaling optimized for…”.

**Have all data underlying the figures and results presented in the manuscript been provided?**

Reviewer #1: None

Reviewer #2: Yes

Reviewer #3: Yes

PLOS authors have the option to publish the peer review history of their article (what does this mean?). If published, this will include your full peer review and any attached files.

Reviewer #1: **Yes: **Hongyan Hao

Reviewer #2: No

Reviewer #3: No

---

## [Decision Letter · Decision Letter 1]

15 Dec 2024

Dear Dr Wang,

We are pleased to inform you that your manuscript entitled "A versatile site-directed gene trap strategy to manipulate targeted gene activity and control gene expression in Caenorhabditis elegans" has been editorially accepted for publication in PLOS Genetics. Congratulations!

Yours sincerely,

Daniel A. Starr

Academic Editor

PLOS Genetics

Quanjiang Ji

Section Editor

PLOS Genetics

Aimée Dudley

Editor-in-Chief

PLOS Genetics

Anne Goriely

Editor-in-Chief

PLOS Genetics

Comments from the reviewers (if applicable):

Reviewer's Responses to Questions

**Comments to the Authors:**

Reviewer #1: The authors have addressed my previous questions and concerns effectively, including adding Figure S6 to support the excretory gland cell expression of aex-2, providing detailed protocols for CRISPR/Cas9 gene-editing for generating gene trap strains, clarifying the HisCl1 experiments, and commenting on the different performance of different bipartite expression systems. Overall, the revisions significantly improved the clarity of the study, and I believe the manuscript is now ready for publication.

Reviewer #2: The authors have adequately answered all my concerns.

Reviewer #3: In this revised manuscript by Khan et al., the authors have provided extensive additional data and have extensively revised the text to thoroughly address previous critiques. The authors have now included key expression pattern, key negative controls, as well as extensive details/discussions about their methodology and elusive results related to LexA GT. Confusing text in figure S4 has also been clarified. Overall, the manuscript contains several interesting and relevant findings that will be of broad interest to a wide audience.

**Have all data underlying the figures and results presented in the manuscript been provided?**

Reviewer #1: Yes

Reviewer #2: Yes

Reviewer #3: Yes

PLOS authors have the option to publish the peer review history of their article (what does this mean?). If published, this will include your full peer review and any attached files.

Reviewer #1: No

Reviewer #2: No

Reviewer #3: **Yes: **Chenshu Liu

**Data Deposition**

http://datadryad.org/submit?journalID=pgenetics&manu=PGENETICS-D-24-01003R1

**Press Queries**

---

## [Editor Report · Acceptance letter]

2 Jan 2025

PGENETICS-D-24-01003R1 

A versatile site-directed gene trap strategy to manipulate gene activity and control gene expression in *Caenorhabditis elegans*

Dear Dr Wang, 

We are pleased to inform you that your manuscript entitled "A versatile site-directed gene trap strategy to manipulate gene activity and control gene expression in *Caenorhabditis elegans*" has been formally accepted for publication in PLOS Genetics! Your manuscript is now with our production department and you will be notified of the publication date in due course.

With kind regards,

Anita Estes

PLOS Genetics

On behalf of:
